# T Cells Subsets in the Immunopathology and Treatment of Sjogren’s Syndrome

**DOI:** 10.3390/biom10111539

**Published:** 2020-11-11

**Authors:** William de Jesús Ríos-Ríos, Sorely Adelina Sosa-Luis, Honorio Torres-Aguilar

**Affiliations:** 1Department of Clinical Immunology Research of Biochemical Sciences Faculty, Universidad Autónoma “Benito Juárez” de Oaxaca, Oaxaca City 68120, Mexico; qfbrioswilliam@hotmail.com; 2Department of Molecular Biomedicine, Centro de Investigación y de Estudios Avanzados del Instituto Politécnico Nacional, Mexico City 07360, Mexico; minna.leluiso22@gmail.com

**Keywords:** Sjogren’s syndrome, T cell subsets, infiltrating T cells, immunomodulatory cytokines, emerging T cells, therapeutic targets

## Abstract

Sjogren’s syndrome (SS) is an autoimmune disease whose pathogenesis is characterized by an exacerbated T cell infiltration in exocrine glands, markedly associated to the inflammatory and detrimental features as well as the disease progression. Several helper T cell subsets sequentially converge at different stages of the ailment, becoming involved in specific pathologic roles. Initially, their activated phenotype endows them with high migratory properties and increased pro-inflammatory cytokine secretion in target tissues. Later, the accumulation of immunomodulatory T cells-derived factors, such as IL-17, IFN-γ, or IL-21, preserve the inflammatory environment. These effects favor strong B cell activation, instigating an extrafollicular antibody response in ectopic lymphoid structures mediated by T follicular helper cells (Tfh) and leading to disease progression. Additionally, the memory effector phenotype of CD8+ T cells present in SS patients suggests that the presence of auto-antigen restricted CD8+ T cells might trigger time-dependent and specific immune responses. Regarding the protective roles of traditional regulatory T cells (Treg), uncertain evidence shows decrease or invariable numbers of circulating and infiltrating cells. Nevertheless, an emerging Treg subset named follicular regulatory T cells (Tfr) seems to play a critical protective role owing to their deficiency that enhances SS development. In this review, the authors summarize the current knowledge of T cells subsets contribution to the SS immunopathology, focusing on the cellular and biomolecular properties allowing them to infiltrate and to harm target tissues, and that simultaneously make them key therapeutic targets for SS treatment.

## 1. Introduction

Sjogren’s syndrome (SS) is a complex, inflammatory, autoimmune disorder characterized by damage to the salivary and lacrimal glands, which may lead to the loss of appropriate tear and saliva production, resulting in symptoms of severe dry eyes and mouth. The pathology in SS may additionally extend from sicca symptoms and complications of mucosal dryness, as a result of exocrine gland involvement, to a systemic disease or even to malignant B cell lymphoproliferation. SS is called “primary” (pSS) when it occurs alone, or “secondary” (sSS) if it is associated to the presence of another autoimmune disease [1]. Current evidence suggests that T cells form a large part of the lymphocytic infiltrated at earlier stages of the disease, involved in tolerance loss to self-antigens and in the secretion of many pro-inflammatory cytokines associated to local inflammation [2,3].

T cells comprise the helper T cell populations (CD4+), which differentiate in several subsets such as Th1, Th2, Th17, regulatory T cells (Treg), and T follicular helper cells (Tfh), as well as CD8+ T cells, also called Cytotoxic T Lymphocytes (CTL) [4]. Increased T cells infiltration into salivary glands (SG) from pSS patients has been evidenced accomplished by decreased levels in periphery blood, supporting the hypothesis that lymphopenia, a frequent finding in pSS patients associated with higher disease activity and increased mortality, might be owed to T cells migration [5].

Both Th1 and Th17 cells subsets infiltrating the SG at an early disease stage have been evidenced by detection of interferon (IFN)-γ and interleukin (IL)-17 respectively, being highly associated with the inflammatory damage [6]. In regard to regulatory T cells (Treg), some studies show conflicting data about their frequency in blood and target organs, displaying uncertain effects. Although circulating Treg cells have not been shown to be significantly decreased with impaired clonal expansion and functionality [7], their physiological and pathological role in SS is unclear yet. Tfh cells are taking special attention for their essential roles in ectopic lymphoid structures (ELS) development in pSS patients, due to their germinal center (GC)-like organization that allows a potent B cell response [8]. On the other hand, CTL have also been implicated into SS pathology, since data from a murine SS model and human biopsies reveal the pathogenic significance of CD8+ T cells in the development and progression of SS in the SG [9]. Strikingly, a novel T cells subset is emerging, namely a kind of regulatory T cell localized in the GC to limit the humoral response, called follicular regulatory T cells (Tfr), which might play a critical protective role, since their deficiency affects the salivary glands with lymphocyte infiltration and antibody deposition in a mouse experimental model of SS [10]. Thus, T cells display several forms of implications in SS pathogenesis, such as an increased infiltration in the target tissue, contributing to the inflammatory microenvironment and leading to the damage of exocrine glands, and even to B lymphoma prevalence through releasing multiple cytokines governing B cells response. Additionally, recent evidence depicting molecular pathways in SS pathogenesis has allowed for the discovery of novel potential targets directed to T cells response. Hence, T cells immunobiology can be selectively inhibited either directly or indirectly by targeting activator and survival factors, as well as other molecules implicated in their pathological roles in SS [6,11,12].

## 2. T Cell: Targeted Shots in Sjogren’s Syndrome

Mechanistically, many factors are largely and recently associated with the involvement of T cells and the detrimental conditions developed in SS patients. Genetic associations have pointed to the aberrant biology of T cells that can carry weight in SS development [13,14]. In pSS, the lymphocytic infiltration of the salivary and lacrimal glands may organize into B and T cell areas, where different T cell subsets may be involved in the pathology by infiltration, secretion of pro-inflammatory cytokine, damage to epithelium, and also by regulating the function of other immunological cells. On the other hand, some biological processes such as autophagy are acquiring a novel interest in the intracellular events that regulate the immunobiology of T cells since it is crucial for their development, proliferation, survival and functions [15]. Autophagy is a mechanism that has been recently involved in the etiology and development of autoimmune/anti-inflammatory disease as systemic lupus erythematosus, inflammatory bowel disease, rheumatoid arthritis, psoriasis and multiple sclerosis [16]. Although there are few evidences depicting the association of autophagy and SS, some studies suggest that dysregulated autophagy could be involved in the aberrant activation of T cells [17,18]. On the other hand, regarding the break of self-tolerance at the T cell level, the activation of autoreactive T cells by commensal microbiota-derived antigens has been explored as molecular mimicry, prompting SS onset [19]. Additionally, a persistent viral infection has been related to the high frequency of CTL in pSS patients [20]. Likewise, T cells hyperactivity related to dysregulated immune checkpoint signaling pathways could play an important role in pSS pathogenesis. The above is owing to a regulatory pathway for T cell activation associated to autoimmune diseases: the axis T cell Ig and ITIM domain (TIGIT) and CD226 (TIGIT/CD226) [21], was overexpressed in T cells from pSS patients. This suggests that this axis might be involved as an unsuccessful negative immune regulation, making it a potential therapeutic target for this disease [22].

CTLA-4/CD28 axis remains the most studied pathway working as a major immune checkpoint regulating T cells activation in SS and other autoimmune diseases [23]. For this reason, abatacept (a humanized cytotoxic T-lymphocyte–associated antigen 4 (CTLA4)–IgG1 fusion protein binding CD80 or CD86 and inhibiting the CD28 co-stimulatory pathway on T cell therapy) has been evaluated for SS treatment [24,25,26]. The focus towards TIGIT/CD226 pathway similarly relies on the CTLA-4/CD28 pathway owing to the TIGIT/CD226 pathway that exerts its immunomodulatory effects by competing for the same ligand. Moreover, additional pathways like PD-1/PD-L and ICOS/ICOSL have also been considered as defective immune checkpoints associated with T cells hyperactivity in SS [23].

Thus, regarding all those aforementioned aspects, here, we discuss the current knowledge of T cells subset involvement in SS, focusing on their properties for the infiltration of exocrine glands and their pathologic mechanisms, as well as the novel emerging factors making them potential therapeutic targets for the treatment of SS.

## 3. Th1/Th17 Cells: Primordial Effectors Coordinating the Inflammatory and Detrimental Environment

Th1 and Th17 cells have long been implicated as crucial mediators at early stages of SS pathology, contributing in shape up the inflammatory microenvironment (Figure 1). Such notion was emphasized by the fact that both pSS-infiltrating activated Th1 and Th17 subsets presented restricted clonal diversities with TCR recognizing common autoantigen unique to pSS [27]. Th1 cells have also been narrowly associated to development and disease severity, owing to their selective infiltration into target tissues, analyzed by the presence of the transcription factor T-bet and IFN-γ [28]. Alternatively, some studies have found an increased expression of predominantly Th1-cytokines, such as IL-2, IL-12, IFN-γ, and IP-10, in liquids like tears and saliva from pSS patients correlating with clinical manifestations [29,30]. Consistent with the underlying mechanism to shape the setting of the inflammatory and detrimental environment, IFN-γ might disrupt tight junction structure in glandular tissue from SS patients [31], indicating that the local cytokine production may contribute to the observed glandular dysfunction. In addition, it is very well known that IFN-γ regulates the activity of certain innate immune cells, whose overactivation may further contribute to SS pathology [32,33,34,35]. On the other hand, TNF-α is another Th1 cytokine associated to SS development in murine model, detected at high levels in serum [36] and exhibiting enhanced expression in saliva fluid and salivary glands from SS patients [30,37]. Although TNF-α can be produced not only by T cells, this cytokine effectively acts on both naïve and effector T cells by regulating their proliferation and survival [38]. Further, TNF-α can mediate inflammatory activity via special effects on structural cells through regulating the reorganization of epithelial junctions and inducing the expression of adhesion molecules, such as E-selectin, facilitating leukocytes adhesion [39]. Alike IFN-γ, the detrimental roles of TNF-α on glandular tissues from SS patients are due to its tight junctions-disrupting effect [31]. Furthermore, TNF-α can impact at the inflammatory microenvironment by stimulating secretion of pro-inflammatory cytokines like IL-8, which in turn increases leukocyte infiltration, particularly neutrophils, whose extracellular traps have been related to SS pathogenesis [40,41]. Additionally, TNF-α induce amphiregulin (AREG) secretion by epidermal cells, a growth factor whose presence has been demonstrated to play critical roles in pro-inflammatory cytokine secretion in SG from SS patients [42,43].

Th1 cells can be attracted at SG by exacerbated secretion of specific chemokines; therefore, knowledge of those mechanisms allowing such migration is essential to elucidate and to treat SS. IL-7 is a cytokine promoting T cells development and survival. Additionally, it may induce expression of several chemokine favoring massive T cells homing towards many tissues [44]. Interesting data shows that Th1 cells may promote development of SS-like autoimmune exocrinopathy in murine model, supported by enhanced expression of CXCR3 ligands in a IL-7-dependent manner [45]. This notion is relevant owing to an increased expression of IL-7 correlates with increased inflammation in SS [46] and the expression of CXCR3 ligands (including CXCL9, CXCL10, and CXCL11) are elevated in SG from SS patients [47].

Regarding TH17 cells involvement, both experimental and clinical data have pointed their crucial role in development and progression of SS by supporting autoreactive B cells responses. pSS patients show increased levels of circulating Th17 cells, as well as in salivary glands correlating with clinical parameter [48,49]. Th17 cells involvement in SS can be also analyzed by measuring their associated cytokines, particularly IL-17. This cytokine has been found significantly augmented in several exocrine glands from pSS patients, correlating with the severity of the pathology [50,51]. Moreover, the Th17/Treg cell imbalance ratio in peripheral blood from SS patients with extra glandular complication, relates the possible role of Th17 cells in the development of systemic manifestations [52]. The ontogenesis of Th17/Treg cells imbalance has recently been implicated to transcriptional regulators. The dysregulated expression of the transcriptional co-activator TAZ (transcriptional coactivator with PDZ (postsynaptic density 65-discs large-zonula occludens 1-binding)) leads to autoimmune diseases such as SS by promoting Th17 differentiation and attenuating Treg development in a mouse model. Further, a higher expression of TAZ has been demonstrated in circulating CD4+ memory T cells from pSS patients [53]. Thereby, the aforementioned statements highlight the interest for depicting underlying intracellular events concerning to Th17/Treg imbalanced driving to development of SS.

Even though low evidence supporting this role until now, IL-17 has been described as a novel Th17 mechanism associated with the development of ELS in SS [54]. Th17 cells are more effective than Th1 cells in supporting B cells responses under autoimmunity conditions like SS, and an increased numbers of circulating RORγ+CD161+CD4+Th17 subset positively correlated with humoral manifestations [48,55]. IL-17 might contribute by modulating the immune response, since this cytokine induces secretion of inflammatory factors such as TNF, IL-6 and IL-1β and promotes immune cells recruitment by inducing secretion of chemokine like IL-8, CXCL9 and ligand for CCR3 receptors, as well as by modulating the integrity of epithelial barrier trough secretion of proteins like metalloproteinase [56,57]. Additionally, there are six IL-17 isotypes, (IL-17A–IL-17F), with IL-17A and IL-17F being the more prevalent isotypes. Available data display that both IL-17A and IL-17F are more associated to SS, since high levels of IL-17A are found in salivary glands from pSS patients [58], and IL-17F was correlated with the humoral and disease activity [59]. On the other hand, IL-22 (another TH17 cytokine) favors B cells recruitment and lymphoid aggregation to form ELS, as well as the production of autoantibodies by inducing the expression of several cytokines such as CXCL12 and CXCL13 [60]. This aspect is relevant owing to higher IL-22 levels correlate with SS parameters such as lower saliva flow, autoantibodies profile of anti-SSB, anti SSA/SSB combined, rheumatoid factor and hypergammaglobulinemia [61]. Thus, current evidence shows how Th17 cytokines have become crucial orchestrators of the aberrant immunity response in SS.

## 4. Th2/Treg Cells: T Cells Subsets with Ambiguous Implications

Th2 cells secrete IL-4, IL-5, and IL-13 as main cytokine signature modulating protective immune responses against helminths and tissue repair. Th2 polarization has better been characterized in the pathogenesis of chronic inflammatory conditions like allergy and asthma [62]. Nevertheless, the paradigm of Th1/Th2 imbalance is a feature of some autoimmune responses. In this line, conflicting data shows no significant differences in frequency and activity of both populations, although the Th1/Th2 ratio is displayed significantly higher in pSS patients compared to controls [63,64]. Therefore, there is not clear evidence showing crucial effects of the Th1/Th2 imbalance leading to ambiguous conclusions. Nevertheless, serious insights about the association between Th2 and SS disease have been linked. Th2 cells mediate effects encouraging B cells response trough their cytokines, essentially by IL-4 action [65]. In selectively extracted lesions of Labial SG (LSG), Th2 molecules are detected in zones inside the GC from pSS patients [28], suggesting that Th2 cells might modulate initial B cells responses. However, the harmful roles of Th2 cells might be owed to invasion into specific zones and disease progression, since Th2 cells localized outside lymphocytes infiltrates are associated with beneficial effects, and normal Th2 cytokine genotypes (IL-4 and IL-13) are associated with a milder form of pSS [66,67]. Furthermore, the increased expression of IL-13 cytokine has been implicated in the pathogenesis of autoimmune disease including SS, regulating several responses of other T cells [68]. In regard to IL-4, elevated levels are present in tears and salivary fluid from pSS patients, correlating with clinical parameters [30,64]. Nevertheless, it is necessary to elucidate the real source for this cytokine, since additional immune and no immune cells might produce it. In contrast, the implication of Th2 cells in disease progression can be approached with the comprehension about the effect of their diminished frequency, since reduced Th2 cells frequency in NOD mice with SS-like disease aggravates the disease [69]. Additionally, the beneficial role of Th2 polarization in SS has also been indirectly associated through its cytokines. Thymic Stromal Lymphopoietin (TSLP), a cytokine involved in promotion of Th2 response, has showed a reduced expression in SG with systemic inflammation in pSS, likely allowing a Th1/Th17 associated pSS pathogenesis [66]. Thus, Th2 cells might perform distinct roles in SS disease, providing special features that promote B cells responses towards disease progression or being a flank of altered regulatory pathways.

Treg cells are special populations performing essential functions for maintaining immunological self-tolerance and homeostasis, exerting their suppressive activity by releasing soluble mediator or through cell-cell contact [70]. Regarding their roles in SS, data from various studies have not launched convincing information about their implications, nor associations between frequency of circulating and infiltrating Treg cells with clinical features. Likewise, conflicting evidences show inflexible amount of Treg in pSS patients, where it might be increased, reduced o invariable, and displaying ambiguity [71]. Nevertheless, the optimums phenotype for Treg cells identification should be considered as an important issue, since some commonly used markers, like CD25, are shared by other immune cells leading to unclear data. Treg detection as CD4+CD25+ FoxP3+ in SG from pSS patients showed an enriched infiltrating population [72,73]. However, their implications are not clear since they might be present as unique Treg cells subset related to different disease states. Furthermore, the frequency of infiltrating peripheral FoxP3+ Treg cells have been found correlating according to lesion severity with the lowest frequency obtained in mild lesion and related with the grade of inflammation [74]. Alternatively, it was recently shown that Treg cells may acquire a pro-inflammatory phenotype in some autoimmune disease, with the loss of their suppressive activity [75]. Data from SS murine model reveal that Treg cells can exhibit Th1-like inflammatory phenotype based on the large amount intracellular IFN- γ production [76]. In contrast, the frequency of Treg cells in peripheral blood of SS patients was shown to be dependent of disease activity, with lower levels when the disease activity increases and the Th17/Treg cells ratio is raised [77]. This effect suggests an abnormal Treg cells expansion and differentiation as demonstrated in SS murine model; revealing that this imbalance may play crucial role underlying the SS pathology [75]. Moreover, a novel Treg cells subpopulation, Helios+FoxP3+ Tregs, is recognized representing a functional stage that provide better effective suppressive activity that those Helios- Treg cell populations [78]. Increased levels of Helios+FoxP3+ cells in peripheral blood of SS patients got inverse correlation with certain clinical parameter, suggesting that this population might be implicated in sustaining suppressive activity [79]. On the other hand, abnormalities in Treg cells immunobiology may be considered as one pathological mechanism in SS. CCR7+ Treg cells frequency is decreased in SG from SS patients and the in-situ analysis of patrolling CCR7+ Treg cells in target organs revealed that an optimums migration is essential for controlling lesions in SS murine model [80]. However, CCR7+ T cells-controlled migration is not limited to Treg cells and findings suggest that CCR7 might play an important role in the development of pSS, since their expression on circulating CD4+ T cells correlated with disease activity but not with damage severity [81]. In addition, it is well established that the IL-2 pathway is critical for Treg cells development and expansion by controlling effector T cells responses [82]. Luo et al. found that, although there were unchanged percentages of Treg cells in periphery and SG from pSS patients in comparison to healthy control, IL-2 levels were reduced, and IL-2/IL-6 ratio was altered. Interesting, they showed that in vitro IL-2 treatment inhibited Th17 generation. This fact suggests a possible IL-2 mediated suppression, since additional IL-2 therapy restored the circulating Th17/Treg cells imbalance in pSS patients [7]. However, IL-2 measuring could be contradictory in this line of reasoning due IL-2 might be secreted by other T cells subsets like Th1, and high levels of IL-2 have been correlated with clinical parameters [30]. Thus, the altered expansion and abnormal phenotype of Treg cells may contribute to the impaired control of the overactivated immune response, and further studies should be performed to elucidate the actual contribution of Treg cells based on their functional capability beside their frequency.

## 5. Tfh Cell: Building the Niche Leading to Progressive Damage

Tfh cells are specialized helper T cells essential for development and activity of GC for aiding B cells response. Tfh cells immunophenotyping is based on sustained surface expression of CXCR5^high^, ICOS, and PD-1, intracellular expression of the essential transcription factor Bcl-6, and secretion of abundant IL-21 amounts, although some variations are dependent on their source and activated state [83]. Tfh cells are increased in peripheral blood of SS patients [83,84], and activated Tfh cells are associated with disease activity and have been considered as biomarker predictors of tissue damage [85,86]. Furthermore, circulating Tfh cells represent a heterogeneous population based on surface marker expression and cytokine production, existing distinct subpopulations acquiring different activities in supporting B cell response in several autoimmune disease [87]. There is also a circulating T cells population related to Tfh cells according to CD4+CXCR5+ expression, but displaying distinct phenotype, renamed “Tfh-like cells”, which have been found increased in pSS patients correlating with clinical parameters [84,88]. For instance, there areTh17-like cells expressing markers for Tfh cells that have been detected significantly increased in pSS patients, correlating with humoral parameters and disease activity [84]. Strikingly, the CCR7^low^PD1^high^ subset is a novel population of circulating Tfh cell that have been linked to Tfh as a partial effector phenotype, and their differentiation was suggested occurring before GC formation [86,89]. Moreover, this novel subset showed to mediate better B cell response than other CD4+CCR5+ T cell populations [89]. In regard to SS, the CCR7^low^PD1^high^ subset was displayed increased in blood from pSS patients and their frequency significantly correlated with disease activity and SG inflammation [86].

PD-1^hi^CXCR5− CD4+ T cells described as peripheral helper T-cells (Tph) lie in the “Tfh-like cells” population, and increased numbers of Tph cells in peripheral blood of pSS patients compared to healthy controls have been observed, but in lower frequency compared to circulating Tfh-cells [25,90], denoting a probable minor implication. However, a recent study showed in SG from SS bearing ELS, that Tph cells are enriched, contributing as main mediator of IL-21 and INF-γ secretion [91]. Further, owing to Tph roles that have been related to limiting the aid for memory B cells [92], their implication might probably be established at the course of clinical manifestations where the B cells response is active. This aspect in turn may endow Tph with the capability to modulate B cells hyperactivity in SS, although more evidence needs to be shown to elucidate their crucial implications.

The main known function of Tfh cells is as crucial elements for GC development and orchestrating several activities during GC reaction for supporting the B cells response. In addition, Tfh cells aids high affinity B cells selection to differentiate into high-affinity antibodies producing cells and memory B cells to trigger ideal responses to T dependent antigens [93]. Mechanistically, the main pathological implication attributed to Tfh cells in SS is to aid self-reactive B cells response. Tfh have been found in SG from pSS patients promoting B cells maturation [94]. Further, Tfh are strongly implicated to ELS formation (a hallmark of SS disease), since increased activated Tfh have been detected in SG with ectopic GC formation [28]. Nevertheless, there are several factors influencing Tfh cells activity in SS pathology. For instance, IL-21 works as an autocrine and paracrine cytokine crucial for lymphocyte differentiation, expansion and survival; and additionally IL-21 support B cells migration and differentiation in secondary lymphoid organs [95]. Notably, IL-21 is increased in serum and SG tissue from pSS patients correlating with disease activity, and IL-21 pathway has been implicated as critical factor in development of SS in both animal and human disease [96,97,98,99]. A recent finding in animal model of SS discovered that a novel mechanism of IL-21 pathway in SS pathology is due to downregulated Pax3-Id3 signaling activity, which suppressed Tfh activation. Such molecules were demonstrated to be reduced in SG from SS patient compared with healthy controls [100]. Interesting data from this study revealed that upregulation of Pax3-Id3 expression or blocking IL-21 receptor reduced Tfh cell frequency and alters SS development. Thus, IL-21 pathway becomes a striking target for therapeutic strategies in SS. Additionally, in the line of SS pathology-influencing Tfh factors, in vitro analysis valuing pSS patients-derived SG epithelial cells revealed that these cells promote naïve T cells differentiation into Tfh in an IL-6 and ICOL dependent manner [97]. This evidence suggests that Tfh may mature at site of inflammation, and IL-6 exhibits crucial roles for Tfh cell generation during humoral response, which would allow focusing on IL-6 as a target to downregulate Tfh cells frequency, since IL-6 is increased in SS [29]. Alternatively, Achaete-scute complex homologue 2 (Ascl2) is a novel molecule that has been reported to induce Tfh cells differentiation. Analysis about the influence of this transcriptional factor in SS murine model releveled their association with abnormal Tfh cell differentiation, since high Ascl2expression is elevated in T cells from mice and SS patients. Moreover, Ascl2 overexpression induces increased CXCR5 expression, with likely implications in elevated Tfh infiltration, and increased IL-21 and Bcl-6 levels [101,102], suggesting that Ascl2 would play an essential role in SS progression by inducing Tfh cell expansion. On the other hand, migratory Tfh cells activity takes attention in progressive B cell response, since CXCL13, the B cell chemoattractant, has recently been linked to lymphocyte infiltration in SS. Tfh cells migrate into B cell follicles in response to gradient CXCL13 expression, which binds to CXCR5, and Tfh cells may also produce CXCL13 [83]. Increased CXCL13 levels are detected in plasma and SG from pSS patients [1,103], and increased serum CXCL13 concentrations have been correlated with aberrant B cell parameters even associated with lymphoma development [104]. Even though there is not enough evidence relating high CXCL13 levels with Tfh, more studies should be taken, since both CXCL13 and CXCR5+ cells play key roles in GC activity [105], and genetic susceptibility of CXCR5 gene have been related to T cells homing to target tissue [1].

Besides their outstanding participation in promoting B cells hyperactivity, T cell-derived cytokines like INF-γ may trigger B cells activation by inducing B cell activation cytokine such as BAFF and APRIL which are elevated in pSS patients. Likewise, upregulated IL-6 levels induced by IFN-γ may in turn be involved in the enhanced plasma cells formation; shaping in this way, an axis with IL-21 contributing with the B cells hyperactivity shown in pSS patients [106]. Parallelly, secretion of high levels of B cell recruiting chemokines whose expression is regulated by T cell-derived proinflammatory cytokine, trigger B cell hyperactivity endowing them with a susceptibility to elicit autoreactive activities into target tissue [107].

## 6. Cytotoxic T Cells

CD8+ T cells, also called cytotoxic T lymphocyte (CTL) are key players of the adaptive immune system against intracellular pathogens and malignant cells. CTL secrete cytokines like IFN-γ and TNF-α, release cytotoxic granules and express high Fas-L levels upon activation to mediate direct cytotoxic cellular activity [108]. Few data are available depicting CTL implications in SS. Strikingly, multiomic analysis concerning to whole blood transcriptomes, serum proteomes and peripheral immunophenotyping reveal that CD8+ T cells are associated with gene signature of pSS disease, mainly related to terminally differentiated effector memory CTL, and such association was also shown in SG from SS patients [109]. A study performed by Narkevicuite et al. [20] displayed decreased CD8+ T cells frequency within circulating lymphocyte population compared to non-autoimmune sicca syndrome group. Further, the frequency of various CTL subsets reveals the existence of altered number of cytotoxic memory/effector T cells. Although this study focused on analyzing associations between lymphocytes changes with persistent viral clearance and its relationship with SS etiopathogenesis, it might suggest that altered frequency of circulating CTL might be related to their high infiltration into damaged tissues. In this line, based on HLA-DR+ expression, a high number of activated CD8+ T cells was found infiltrating LSG [3], and based on CD69 and CD103 expression, CD8+ T cells with a tissue-resident memory phenotype (CD8+TRM cells) were found as dominant infiltrating cells, adjacent to salivary duct epithelial cells [9].

On the other hand, in a murine model of SS disease, infiltrating cytotoxic CD8+ T cells driving pathologic implications has been documented. Activated CD8+ T cell are present in salivary and submandibular glands from mice with the capability to produce high levels of IFN-γ [9,110]. Adoptive transference of CD8+ T cell with cytotoxic phenotype may drive specific inflammation in lacrimal gland in NOD-SCID mice [110]. Furthermore, CD8+ T cell-specific depletion abrogates SS features development, and blocking IFN-γ production decreases T cells infiltration [9].

Data from several studies have demonstrated higher levels of IFN-γ and TNF-α in blood, fluids and target glands from pSS patients, getting the possibility that this cytokines could be produced by CTL, since their production is not limited to Th1 cells [30,111]. CD8+ T cell recruitment to target tissue is dependent of chemokines pathways. Similar to Th1 cells, CD8+ T cells express CXCR3, and increased levels of CXCR3 ligands have been found in saliva, tears, and SG from pSS patients [112,113]. A recent study showed that CXCR3 blocking impedes development of clinical signals in SS murine model reducing percentage of infiltrating CXCR3+ CD8+ T cells and TNF-α expression [47]. For the aforementioned reasons, even though there is little evidence depicting CTL implications in SS pathology, available data suggests that infiltrating activated CD8+ T cells might cooperate to uphold the inflammatory environment. Moreover, the fact that SG-infiltrating CD8+ T cells display a memory phenotype suggests that CTL might represent self-reactive T cells triggering autoimmune responses and tissue damage.

## 7. Emerging T Cells

Angiogenic T cells (Tang) are a new T cells population characterized by CD3, CD31 and CXCR4 expression, cooperating to shape up Endothelial Progenitor Cells (EPC) colony and to induce endothelial cells proliferation by secreting proangiogenic factors like vascular endothelial growth factor (VEGF), IL-8 and IL-17 [114], thus cooperating in new blood vessels formation. Uncontrolled neovascularization has been proposed to take place in SS extraglandular manifestations, contributing to inflammation and exacerbation [115,116]. Higher Tang cells numbers have been recently reported in peripheral blood from pSS patients correlating with disease activity and with their capability to produce high IL-17 levels, as well as increased in SG in comparison to non-pSS sicca controls [117]. Some authors suggest that Tang cells might be a novel T cell related to glandular neo-angiogenesis and inflammation leading to endothelial dysfunction observed as extraglandular manifestations in pSS patients. Even though more evidence is necessary to reinforce Tang cell pathologic roles in SS, these cells have been widely associated with endothelial dysfunction in other autoimmune disease [115,118,119].

Follicular regulatory T cells (Tfr) (CD4+CXCR5^high^PD-1^high^Foxp3+Blimp-1+) were recently discovered as a Treg cells subset with implications in regulatory pathways on SS. Tfr are important regulatory cells expressing potentially autoreactive TCRs to preserve the immune tolerance by regulating GC reaction and controlling both Tfh and B cell responses [120,121]. Several studies have investigated the ratio of circulating Tfh/Tfr cell subsets in SS, as well as its association with abnormal B cell response. Altered Tfh/Tfr ratio is found in peripheral blood and SG correlating with ELS presence and strong B cells infiltration [85]. However, these regulatory T cells have been also found increased in peripheral blood, correlating with humoral responses in patients with high autoantibody titers [86]. Although increased level of circulating Tfr cells have been found, data suggest that Tfr cells have not optimal capability to suppress B cells response [122]. In addition, Tfr cells deficiency enhanced SS development, as demonstrated in murine experimental model [10].

Double negative (DN) TCRαβ+CD3+ T cells are characterized by bringing neither CD4 nor CD8 molecules expression, comprising a small and heterogeneous subset considered as terminally differentiated effector peripheral T cells. Nevertheless, under inflammatory conditions DN T cells levels increase in peripheral blood and target tissues, exhibiting distinct effector phenotypes [123]. DN T cells have been described as expanded in peripheral blood from pSS patients, acting as major IL-17 producers, accumulated in SG and associated to disease activity. Interestingly, it was shown that IL-17-producing DN T cells from these patients were resistant to dexamethasone [124]. In addition, findings suggest that DN T cells are actively involved by leading to glandular dysfunction and tissue damage, and may play a key role promoting ectopic lymphomagenesis development in pSS [125].

## 8. Molecular Therapeutic Targets in T Cells

The discovery of emerging biomolecules regulating T cell immunobiology and their subsequent involvement in both the onset and progression of SS, has made T cells key targets for designing therapeutic strategies to SS treatment.

With the goal of inhibiting T cell activation, several agents have been tested. One of the best selective compounds is the biological agent “Abatacept”, a humanized cytotoxic T lymphocyte associated antigen 4 (CTLA4)-IgG1 fusion protein binding CD80 or CD86 on Antigen Presenting Cells (APC) like B cells, inhibiting the costimulatory CD28 pathway on T cells. Some open-label studies evaluating the primary endpoint and performed by several medical centers suggest efficacy and safety of abatacept for pSS treatment. Histologic evaluation as well as serologic and clinical changes in response to abatacept display reduction of infiltrating lymphocytes (but also the frequency of Treg cells), increased circulating lymphocytes, high saliva flow along with reduction of glandular inflammation, reductions of disease activity, and prolonged remission [126,127,128]. However, data from further studies suggest that their efficacy and safety get contradictory results [24,25]. Therefore, additional studies regarding clinical parameter and biological features may be encompassed. Mechanically, abatacept therapeutic has displayed to disrupt Tfh cells-dependent B cell hyperactivity evidenced by reduced frequency of circulating Tfh cells and ICOS expression significantly correlating with lower disease activity and reduced plasmablast frequency. Such events may affect T cell–B cell interaction [26]. In this line, analysis of histopathological changes in parotid gland tissue from pSS patients displayed that abatacept may reduce GC in in salivary gland tissue as main feature [129]. As aforementioned, studies about the role of Treg on SS patients have launched conflicting evidences showing increased, reduced o invariable amount and displaying ambiguous implications [72]. However, IL-2 therapy has been used to treat pSS patients, exploring its short-term effects on T cells. A low dose therapy with IL-2, restored the Th17/Treg cells balance by augmenting CD4+ Treg cells numbers [77]. Nevertheless, there is a limited repertoire of T cell-targeted therapeutic agents evaluated in clinical trials; the existent ones display controversial results, coursing different phases, whose status are either finished, recruiting or with inability to meet objective protocols leading to premature termination (reviewed by Felten [11], Mavragani [90]).

According to experimental studies, a recent report for ALPN-101, a fusion protein simultaneously inhibiting CD28 and ICOS costimulatory pathways, showed better suppression in comparison to abatacept, by reducing release of pro-inflammatory cytokine such as TNF-α, IFN-γ and IL-6 in peripheral blood mononuclear cells from SS patients [130]. On the other hand, representative experimental strategies used in animal models have delivered promising data targeting T cell immunobiology. For instance, inhibition of impaired autophagy reduces pathological features of the disease [18]. Administration of a neutralizing anti-TNF-α antibody during the stage prior to SS onset in NOD mice significantly improved salivary secretion, decreased the number of leukocyte foci, and reduced T and B cells numbers as well as T-bet protein level in submandibular glands, suggesting a decrease in Th1 and CTL cells infiltration [36]. Blocking IL-21 pathway in a condition where the negative regulator mechanism fails, has been shown ameliorating SS-like diseases manifestations [99,100]. As described above, T cells hyperactive migration is essential for pathological establishment in gland to coordinate the onset and progression of SS. Blocking IL-7 receptor signaling has arisen as a promising mechanism for SS treatment because it disrupts salivary gland inflammation, rescues the secretory function, and reduces leukocyte infiltration and antibody production [45,131]. Likewise, CXCR3 blocking interrupts the development of key pathogenic signals, reducing the percentages of infiltrating CXCR3+CD8+ T cell and the secretion of proinflammatory cytokines [47]. Thus, several factors are arising as potential biomolecular targets (Table 1; Figure 2) for preclinical and clinical research, which are critical to regulate T cell immunobiology in SS pathology.

## 9. Conclusions

T cell functions are essential elements for the onset and development of SS pathology. Such roles are crucial for shaping the cellular environment leading to disease progression. The activation and migratory state of different T cell populations provide functional hyperactivity into exocrine glands. Th1 and CTL cells take key roles triggering an inflammatory and detrimental environment by secreting high levels of IFN-γ and TNF-α. Meanwhile, Th17 cells coordinate the connection between early phases with the evolution of pathology. Altogether these elements sustain the inflammatory environment and allow the establishment of required elements for humoral response through Th17 cytokines secretion. Simultaneously, Tfh cells regulate ELS and GC development, allowing the B cells response to continue, where Th2 cells likely participate at an early stage. Treg cells implications are ambiguous, and thus more studies about their role need to be performed, mainly focusing on their functional state. Strikingly, emerging T cell subsets are arising as interesting cellular mechanisms owing to their associations with clinical parameters. Thus, T cells have become logical and key targets for SS therapeutics. Several agents are under preclinical and clinical investigation. However, studies have presented conflicting data and the currently available repertoire of therapeutic agents is limited towards those used for another autoimmune disease. Therefore, the comprehensive understanding of T cell involvement in SS pathology will allow for the discovery of more biomolecular targets crucial for specific treatments for SS.

## Figures and Tables

**Figure 1 biomolecules-10-01539-f001:**
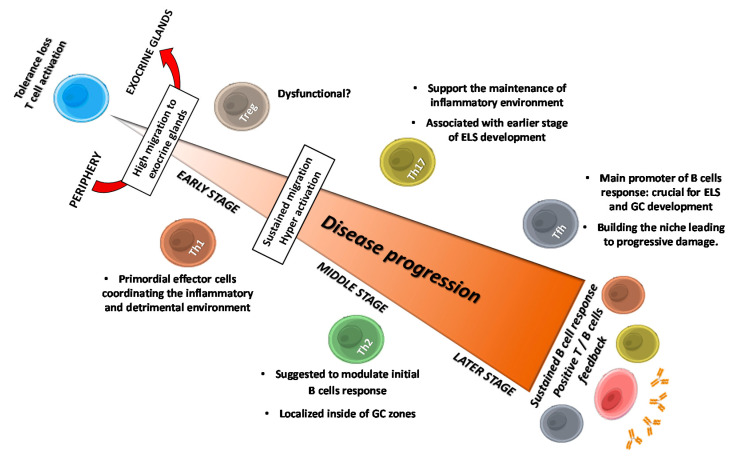
Sequential roles of T cells in SS progression: flares of main implications. Tolerance loss drives T cell activation and migration to exocrine glands, where they provide several factor that make the auspicious environment for the pathology establishment: Th1 and Th17 cells contribute for inflammatory and detrimental environment by secreting cytokines like IFN-γ, TNF-α and IL-17, triggering sustained T cells migration and hyperactivation. Th2 cells are suggested to modulate early B cells response, and to contribute with Th17 cells in initial ESL development, allowing Tfh cells to establish and to provide the necessary factors to trigger B cells response; thus, leading to progressive tissue damage. Treg cells dysfunctionality and low frequency are suggested to contribute to periphery loss tolerance. Positive T and B cells feedback trough intercellular contact and soluble factor allow a sustained B cells response and the onset of clinical features. GC, Germinal Center; ELS, Ectopic Lymphoid Structures.

**Figure 2 biomolecules-10-01539-f002:**
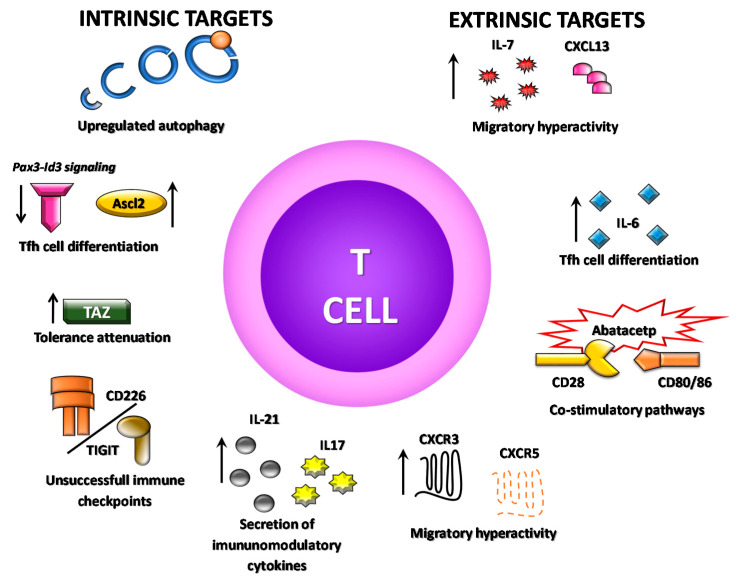
Schematic representation of the several potential T cells alternative targets for SS therapeutic. Novel biomolecules are arising as key factor for controlling several aspects of T cells immunobiology, encompassing those directed towards their activation and migratory sate, as well as those implicated in modulating their inflammatory phenotype.

**Table 1 biomolecules-10-01539-t001:** Potential molecular alternative targets for SS preclinical/clinical research.

Target	Function	Status on SS	Active T Cell Subset in SS	References
Autophagy	Development, survival, and proliferation of T cell	Upregulated on SG	CD4+	[17,18]
CD226/TIGIT axis	Immune checkpoints pathway	Unsuccessful negative regulation on circulating T cells	CD4+, CD8+	[22]
IL-17	Immunomodulatory and pro-inflammatory. Associated with ELS. Support B cells response	Increased in blood, saliva and, lacrimal fluid and glands	Th17, Th1, Tfh	[49,55]
IL-6	Immunomodulatory and pro-inflammatory. Supports Tfh cell generation.			[97]
IL-21	Immunomodulatory. Supports ELS and GC development. Induces B cells migration and survival			[99,100]
IL-7	Induces the expression of several chemokine favoring massive T-cell homing	Increased expression correlating with inflammation in SG	Th1, CTL	[45,131]
CXCR3	Chemokine receptor	Elevated in saliva, tears and SG	Th1, CTL	[47]
CXCL13	Ligand for CXCR5. Chemotaxis	Elevated in plasma and SG	Tfh	[2,103]
TAZ	Promotes Th17 differentiation and attenuates Treg development	Higher expression in circulating CD4+ memory T cells	Th17, Treg	[53]
Pax3-Id3 signaling	Transcriptional regulator of effector Tfh cell activation	Downregulated activity in SG	Tfh	[100]
Ascl2	Transcriptional factor inducing abnormal Tfh cell differentiation	Increased expression in SS patients and mice	Tfh	[101,102]

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
