# Peer review of "T Cells Subsets in the Immunopathology and Treatment of Sjogren’s Syndrome"

_biomolecules, 2020, doi:10.3390/biom10111539_

Round 1

Reviewer 1 Report

Different T cell subsets including effector Tfh, Th17 and Th22 cells; regulatory T cells (Treg) and T follicular helper cells (Tfh) and Cytotoxic CD8+ T cells present complex interacting functions are essential elements for the onset and development of Sjogren Syndrome (SS) pathophysiology. T cells of the lymphocytic infiltrated into salivary glands involved in tolerance loss to self-antigens lead to inflammatory and detrimental environment with production of several pro-inflammatory cytokines chemokines and miRNA have been proved to participate to the development of systemic manifestation. This review already has excellent and extensive introduction of distinctive T cell subsets with their functions and potential therapeutic targets for the treatment of SS.

Suggestion:

1 The more detail pathogenic role of autoreactive effector T cells could promoting B-cell hyperactivity that B-cell-centric interacting with Tfh in SS.

2 What are the implications of pathogenic peripheral-helper T-cells (Tph) in SS.

3.What are the more dysregulated immune checkpoint signaling pathways of T cells rather than focus on axis of T cell Ig and ITIM domain (TIGIT) and CD226 (TIGIT/CD226)

Author Response

Different T cell subsets including effector Tfh, Th17 and Th22 cells; regulatory T cells (Treg) and T follicular helper cells (Tfh) and Cytotoxic CD8+ T cells present complex interacting functions are essential elements for the onset and development of Sjogren Syndrome (SS) pathophysiology. T cells of the lymphocytic infiltrated into salivary glands involved in tolerance loss to self-antigens lead to inflammatory and detrimental environment with production of several pro-inflammatory cytokines chemokines and miRNA have been proved to participate to the development of systemic manifestation. This review already has excellent and extensive introduction of distinctive T cell subsets with their functions and potential therapeutic targets for the treatment of SS.

Suggestion:

  1. The more detail pathogenic role of autoreactive effector T cells could promoting B-cell hyperactivity that B-cell-centric interacting with Tfh in SS.

ANSWER: Following up the pertinent reviewer suggestion, the next paragraph and the corresponding references were included. (highlighted in red letters in lines 322-329)

Besides their outstanding participation in promoting B cells hyperactivity, T cell-derived cytokines like INF-γ may trigger B cells activation by inducing B cell activation cytokine such as BAFF and APRIL which are elevated in pSS patients. Likewise, upregulated IL-6 levels induced by IFN-γ may in turn be involved in the enhanced plasma cells formation; shaping in this way, an axis with IL-21 contributing with the B cells hyperactivity shown in pSS patients [99]. Parallelly, secretion of high levels of B cell recruiting chemokines whose expression is regulated by T cell-derived proinflammatory cytokine, trigger B cell hyperactivity endowing them with susceptibility to elicit autoreactive activities into target tissue [112].

  1. What are the implications of pathogenic peripheral-helper T-cells (Tph) in SS.

ANSWER: Author thanks for the reviewer accurate observation about the recently described Tph taking place in SS immunopathology. Attending this suggestion, the next paragraph and the corresponding references were included. (highlighted in red letters in lines 273-281)

PD-1hiCXCR5 CD4+ T cells described as Peripheral-helper T-cells (Tph) lay into “Tfh-like cells” population, increased numbers of Tph cells in peripheral blood of pSS patients compared to healthy controls have been observed, but in lower frequency compared to circulating Tfh-cells [25], denoting a probable minor implication. However, a recent study showed in SG from SS bearing ELS, that Tph cells are enriched, contributing as main mediator of IL-21 and INF-γ secretion [87]. Besides, owing to Tph roles have been related limiting the aid for memory B cells [88], their implication probably might be established at the course of clinical manifestations where B cells response is active. This aspect in turn may endow Tph with the capability to modulate B cells hyperactivity in SS, although more evidence needs to be shown to elucidate their crucial implications.

  1. What are the more dysregulated immune checkpoint signaling pathways of T cells rather than focus on axis of T cell Ig and ITIM domain (TIGIT) and CD226 (TIGIT/CD226).

ANSWER: Attending the reviewer question the next paragraph and the corresponding reference were included. (highlighted in red letters in lines 95-103)

CTLA-4/CD28 axis remains the most studied pathway working as a major immune checkpoint regulating T cells activation in SS and other autoimmune diseases [23]. For this reason, abatacept (a humanized cytotoxic T-lymphocyte–associated antigen 4 (CTLA4)- IgG1 fusion protein binding CD80 or CD86 and inhibiting the CD28 co-stimulatory pathway on T cell therapy) has been evaluated for SS treatment [24-25]. The focus towards TIGIT/CD226 pathway similarly relies on the CTLA-4/CD28 pathway owing to the TIGIT/CD226 pathway exerts its immunomodulatory effects by competing for the same ligand. Moreover, additional pathways like PD-1/PD-L and ICOS/ICOSL also has been considered as defective immune checkpoint associated with T cells hyperactivity in SS [23]

ADDITIONAL INCLUDED REFERENCES

  1. Kroese, F.G.M.; Andulahad, W.H.; Haacke, E, Bos, N.A.; Vissink, A.; Bootsma, H. B-cell hyperactivity in primary Sjogren`s syndrome. Expert Rev. Clin. Immunol. 2014,10, 483-499, doi: 10.1586/1744666X.2014.891439.
  2. Pontarini, E., Murray-Brown, W.J; Croia, C.; Lucchesis, D., Conway, J.; Rivellese, F.; Fossati-Jimack, L.; Astorri, E.; Prediletto, E.; Corsiero, E., et al. Unique expansion of IL-21+ Tfh and Tph cells under control of ICOS identifies Sjögren’s syndrome with ectopic germinal centres and MALT lymphoma. Ann. Rheum. Dis. 2020, 0, 1-12, doi: :10.1136/annrheumdis-2020-217646.
  3. Yoshitomi, H.; Ueno, H. Shared and distinct roles of T peripheral helper and T follicular helper cells in human disease. Cell. Mol. Immuno.2020, doi: 10.1038/s41423-020-00529-z.
  4. Ceeraz, S.; Nowak, E.C.; Burns, C.M.; Noelle, R.J. Immune checkpoint receptors in regulating immune reactivity in rheumatic disease. Arthitis Res Ther. 2014,16,469, doi: 10.1186/s13075-014-0469-1

Reviewer 2 Report

The manuscript entitled " T cells subsets in the immunopathology and treatment of Sjogren’s syndrome " by de Jesús Ríos-Ríos et al., is a review of the literature in regards to the current knowledge of T cells subsets contribution to the onset and development of Sjogren’s syndrome immunopathology and of some therapeutic agents under preclinical and clinical investigation. I found the review comprehensive and extremely well detailed. 

Author Response

The manuscript entitled " T cells subsets in the immunopathology and treatment of Sjogren’s syndrome " by de Jesús Ríos-Ríos et al., is a review of the literature in regards to the current knowledge of T cells subsets contribution to the onset and development of Sjogren’s syndrome immunopathology and of some therapeutic agents under preclinical and clinical investigation. I found the review comprehensive and extremely well detailed.

ANSWER: The authors appreciate the motivating words of the reviewer and have considered them to improve the quality of the manuscript.
